# Decreasing Significance of Early Allograft Dysfunction with Rising Use of Nonconventional Donors

**DOI:** 10.3390/medicina58060821

**Published:** 2022-06-17

**Authors:** Stephanie Ohara, Elizabeth Macdonough, Lena Egbert, Abigail Brooks, Blanca Lizaola-Mayo, Amit K. Mathur, Bashar Aqel, Kunam S. Reddy, Caroline C. Jadlowiec

**Affiliations:** 1Division of Surgery, Valleywise Health Medical Center, Creighton University, Phoenix, AZ 85008, USA; stephanie.ohara@valleywisehealth.org; 2Division of Gastroenterology and Hepatology, Mayo Clinic, Phoenix, AZ 85054, USA; macdonough.elizabeth@mayo.edu (E.M.); lizaola-mayo.blanca@mayo.edu (B.L.-M.); aqel.bashar@mayo.edu (B.A.); 3Mayo Clinic Alix School of Medicine, Scottsdale, AZ 85259, USA; egbert.lena@mayo.edu; 4School of Medicine, Tel Aviv University, Tel Aviv-Yafo 6997801, Israel; abigail.brooks57@gmail.com; 5Division of Transplant Surgery, Department of Surgery, Mayo Clinic, Phoenix, AZ 85054, USA; mathur.amit@mayo.edu (A.K.M.); reddy.kunam@mayo.edu (K.S.R.)

**Keywords:** donation after circulatory death, marginal donor, organ shortage, deceased donor, graft type, donor pool

## Abstract

*Background and Objectives:* Early allograft dysfunction (EAD) is considered a surrogate marker for adverse post-liver transplant (LT) outcomes. With the increasing use of nonconventional donors, EAD has become a more frequent occurrence. Given this background, we aimed to assess the prevalence and impact of EAD in an updated cohort inclusive of both conventional and nonconventional liver allografts. *Materials and Methods:* Perioperative and one-year outcomes were assessed for a total of 611 LT recipients with and without EAD from Mayo Clinic Arizona. EAD was defined as the presence of one or more of the following: bilirubin > 10 mg/dL on day 7, INR > 1.6 on day 7, or ALT and/or AST > 2000 IU/L within the first 7 days of LT. *Results:* Within this cohort, 31.8% of grafts (*n* = 194) came from donation after circulatory death (DCD) donors, 17.7% (*n* = 108) were nationally shared, 16.4% (*n* = 100) were allocated as post-cross clamp, and 8.7% contained moderate steatosis. EAD was observed in 52.2% (*n* = 321) of grafts in the study cohort (79% in DCD grafts and 40% in DBD grafts). EAD grafts had higher donor risk index (DRI) scores (1.9 vs. 1.6, *p* < 0.0001), were more likely to come from DCD donors (48% vs. 13.8%, *p* < 0.0001), were regionally allocated (*p* = 0.003), and had higher cold ischemia times (median 6.0 vs. 5.5 h, *p* = 0.001). Primary nonfunction events were rare in both groups (1.3% vs. 0.3%, *p* = 0.22). Post-LT acute kidney injury occurred at a similar frequency in recipients with and without EAD (43.6% vs. 30.3%, *p* = 0.41), and there were no differences in ICU (median 2 vs. 1 day, *p* = 0.60) or hospital (6 vs. 5 days, *p* = 0.24) length of stay. For DCD grafts, the rate of ischemic cholangiopathy was similar in the two groups (14.9% EAD vs. 17.5% no EAD, *p* = 0.69). One-year patient survival for grafts with and without EAD was 96.0% and 94.1% (HR 1.2, 95% CI 0.7–1.8; *p* = 0.54); one-year graft survival was 92.5% and 92.1% (HR 1.0, 95% CI 0.7–1.5; *p* = 0.88). *Conclusions:* In this cohort, EAD occurred in 52% of grafts. The occurrence of EAD, however, did not portend inferior outcomes. Compared to those without EAD, recipients with EAD had similar post-operative outcomes, as well as one-year patient and graft survival. EAD should be managed supportively and should not be viewed as a deterrent to utilization of non-ideal grafts.

## 1. Introduction

With the growing number of people awaiting liver transplantation (LT), considerable efforts have been made by the transplant community to expand the available donor pool [1,2,3]. This effort has resulted in the use of an increasing number of donations after circulatory death (DCD), in donors being older, in late allocation, and in steatotic allografts [2,4,5,6,7]. Concerns regarding patient and graft survival coupled with perioperative complications, including ischemic cholangiopathy and increased resource utilization, continue to be barriers limiting more widespread use of such donors [1,8,9,10]. Early allograft dysfunction (EAD) has been shown to correlate with inferior post-LT outcomes [11,12,13,14,15,16]. The prevalence of EAD varies anywhere from 20–30% for standard donation after brain death (DBD) donors to as high as 80% for donation after circulatory death (DCD) donors [4,11,12,13,14,15,16,17]. Along with increasing use of nonconventional, often termed marginal liver allografts, there have also been parallel improvements in surgical technique, perioperative management, and outcomes [4,5,18,19,20]. Causes of EAD are likely multifactorial and dependent on unique characteristics of individual liver allografts, intraoperative events, and recipient demographics [17,21,22,23]. In recent years, EAD has become a common occurrence at our center, which has also paralleled our increasing use of grafts from nonconventional donors. Given this background, the aims of this study were to assess the prevalence and impact of EAD in an updated cohort inclusive of DBD, DCD, and other less conventional liver allografts.

## 2. Materials and Methods

This was a single center retrospective study of patients who underwent LT at Mayo Clinic Arizona between 2015 and 2020. Patients who received a multivisceral transplant or a living donor liver transplant were excluded. Status 1A LT recipients were also excluded. This study was approved by the Mayo Clinic Institutional Review Board, and the need for written consent was waived (ID 20-006586). Two groups were compared: grafts without EAD (Group 1) and grafts with EAD (Group 2). Post-operative outcomes, including ICU and hospital length of stay, development of acute kidney injury (AKI), primary nonfunction (PNF), ischemic cholangiopathy, and one-year patient and graft survival were assessed.

EAD was defined as the presence of one or more of the following: bilirubin > 10 mg/dL on post-LT day 7, international normalized ratio (INR) > 1.6 on day 7, or aspartate aminotransferase (AST)/alanine aminotransferase (ALT) > 2000 IU/L within the first 7 days [11]. The donor risk index (DRI) was calculated using the variables of donor age, cause of death, race, DCD status, partial/split live graft, height, organ location, and cold ischemia time (CIT), as previously defined [3]. AKI was defined using the Kidney Disease Improving Global Outcomes guidelines by an increase in serum creatinine > 0.3 mg/dL or 1.5–1.9 times baseline [20]. Additional definitions used included PNF; early graft failure requiring re-transplantation; occurrence within 7 days of transplant; the absence of technical or immunologic problems; post-cross clamp offers and liver offers allocated to our center after aortic cross clamp at the recovery center; moderate-to-severe graft steatosis; the presence of greater than 30% macrovesicular steatosis on post-reperfusion liver biopsy read at our institution; donor warm ischemia time (dWIT); the time from withdrawal of DCD donor support to aortic flushing with preservation solution; ischemic cholangiopathy; and the presence of non-anastomotic extra- or intrahepatic bile duct strictures in the absence of hepatic artery stenosis or hepatic artery thrombosis identified with imaging [4].

Time-zero formalin-fixed protocol post-reperfusion liver biopsies were obtained for most recipients. Patients received methylprednisolone induction and were then started on triple drug immunosuppression (tacrolimus, prednisone, and mycophenolate mofetil). Maintenance immunosuppression was reduced to tacrolimus monotherapy at 3–4 months post-transplant unless otherwise indicated, with trough levels maintained at 4–7 ng/mL.

### Statistical Methods

Continuous variables were described using means and standard deviations; means and standard deviations, and medians; or medians and interquartile ranges (IQRs). Categorical variables were described using counts and percentages. Baseline and post-transplant characteristics were analyzed using *t*-tests and Chi square analysis. A survival analysis was performed using the Kaplan—Meier method. A multivariable regression analysis was completed, looking at risk for graft loss at 1 year post-LT. Data were analyzed using GraphPad Prism 9.3.1 (2021 GraphPad Software, Inc., San Diego, CA, USA).

## 3. Results

In total, six-hundred and eleven LTs occurred during this period. Within this cohort, 31.8% of grafts (*n* = 194) came from DCD donors, 17.7% (*n* = 108) were nationally shared, 16.4% (*n* = 100) were allocated as post-cross clamp, and 8.7% contained moderate steatosis. Post-LT, EAD was observed in 52.5% (*n* = 321) of recipients.

The recipient and donor characteristics for the two groups are shown in Table 1. Recipients in Group 2 had lower MELD scores (21 vs. 23, *p* = 0.0002) and were less likely to be undergoing re-transplantation (*p* = 0.003). Portal vein thrombosis (PVT, *p* = 0.17) and transjugular intrahepatic portosystemic shunt (TIPS, *p* = 0.17) were observed at low frequencies in both groups. Donors in the EAD group had higher DRI scores (1.9 vs. 1.6, *p* < 0.0001) and BMI (30.0 kg/m^2^ vs. 26.8 kg/m^2^, *p* < 0.0001). Liver allografts with EAD were more likely to be regionally allocated (61.7%, *p* < 0.0001) or to come from DCD donors (48.0%, *p* < 0.0001). There were no differences in dWIT for DCD grafts in Groups 1 and 2 (*p* = 0.53). Grafts in Group 2 had higher CIT (6.0 vs. 5.5 h, *p* = 0.001). Post-cross clamp allocation did not increase the prevalence of EAD (14.8% vs. EAD 17.8%, *p* = 0.33).

Time-zero post-reperfusion liver biopsies were obtained for 81.4% (*n* = 236) of grafts in Group 1 (*n* = 277) and 86.3% (*n* = 277) of grafts in Group 2 (*p* = 0.10). A biopsy without histologic abnormalities was observed in 88.6% of grafts in Group 1 and 82.7% of grafts in Group 2 (*p* = 0.06). Grafts with EAD were more likely to have moderate steatosis (15.2% vs. 8.1%, *p* = 0.01). Hepatocellular necrosis was uncommonly observed in either group (2.1% vs. 2.2%, *p* = 0.97).

Post-LT outcomes are shown in Table 2. There were no observed differences in intensive care unit (ICU, 2 days vs. 1 day, *p* = 0.60) or hospital (6 days vs. 5 days, *p* = 0.24) length of stay (LOS) between LT recipients in Groups 1 and 2. Most grafts with EAD (89.1%) exhibited only one EAD criterion (AST > 2000 U/L during first week post-LT week). Post-LT AKI occurred at a similar frequency in Group 1 and 2 recipients (40.3% vs. 43.6%, *p* = 0.41). PNF events were rare and did not differ between the two groups (*p* = 0.22). Need for early reoperation (within 30 days of LT) occurred in 23.1% of all recipients (20.0% in Group 1 vs. 25.9% in Group 2, *p* = 0.09). Ischemic cholangiopathy occurred in 14.9% (*n* = 23) of Group 2 DCD grafts and 17.5% (*n* = 7) of Group 1 DCD grafts (*p* = 0.69). There were no differences in the types of ischemic cholangiopathy observed (minor, confluence dominant, multifocal progressive, and diffuse necrosis) between grafts in Groups 1 and 2 (*p* = 0.49).

Laboratory studies at one week, six weeks, and one-year post-LT are shown in Table 3. At one week, total bilirubin (*p* < 0.0001), AST (*p* = 0.03), ALT (*p* < 0.0001), and alkaline phosphatase (*p* = 0.01) were all higher in Group 2 grafts. Those differences remained present at 6 weeks post-LT (total bilirubin *p* = 0.004; ALT *p* = 0.05; AST *p* = 0.001; alkaline phosphatase *p* < 0.0001). At one year, there were no differences in total bilirubin *p* = 0.29; ALT *p* = 0.42; AST *p* = 0.18; alkaline phosphatase *p* = 0.08 between graft Groups 1 and 2.

There were no differences in patient (HR 1.2, 95% CI 0.7–1.8; *p* = 0.54) or graft (HR 1.0, 95% CI 0.7–1.5; *p* = 0.88) survival (Figure 1). Median follow-up was 3.6 years (IQR 2.4–5.1) in Group 1 and 4.1 years (IQR 2.7–5.6) for Group 2. One-year patient survival for grafts with and without EAD was 95.9% and 94.7%; one-year graft survival was 92.8% and 92.1%. In a multivariate model, after accounting for differences in graft type and CIT, EAD was not associated with graft loss at 1 year post-LT (OR 1.1, 95% CI 0.6–2.1).

### Graft Type Subgroup Analysis

In comparing DBD (*n* = 417) and DCD (*n* = 194) grafts with and without EAD, EAD was observed in 79.4% (*n* = 154) of DCD grafts compared to 40.1% (*n* = 167) of DBD grafts (Table 4). Recipients of DBD and DCD grafts with EAD had lower MELD scores (*p* = 0.001). Malignancy as an indication for LT was more commonly observed in recipients of DBD grafts with EAD (*p* = 0.001). Donor DRI scores were higher in DCD grafts with (2.3 ± 0.3) and without EAD compared to DBD grafts (*p* < 0.0001). Donor BMI was higher in DBD grafts with EAD (*p* < 0.001). Regional allocation was more commonly observed for DCD grafts with EAD (*p* < 0.0001). The largest proportion of nationally allocated grafts was noted in the DBD EAD cohort (*p* < 0.0001). Moderate steatosis was more common in DBD grafts with EAD (21.6% vs. 4.6%, *p* < 0.0001). CIT was shorter in DCD grafts with and without EAD (*p* < 0.001); DBD grafts with EAD had the longest CIT (*p* < 0.001). There were no differences in ICU (*p* = 0.37) and hospital length of stay (*p* = 0.12) between the subgroups. PNF events were more common in DBD grafts with EAD (*n* = 4, *p* = 0.02). The characteristics of the four grafts with PNF are shown in Table 5. When comparing DBD and DCD grafts with and without EAD, there were no differences in patient (*p* = 0.18) or graft (*p* = 0.36) survival.

## 4. Discussion

With ongoing efforts to increase the utilization of nonconventional donor organs, there has been an observed increase in the frequency of EAD [4,22,23]. Historically, the occurrence of EAD has strongly correlated with poor post-transplant outcomes in published reports using conventional DBD grafts [11,12,13]. Despite data suggesting inferiority, the causes of EAD are likely multifactorial and dependent on the unique characteristics of individual liver allografts and recipients [17,21,22,23]. In recent years, EAD has become a common occurrence at our center along with increased use of grafts from nonconventional donors. Given this background, the aims of this study were to assess the prevalence and impact of EAD in an updated cohort inclusive of DBD, DCD, and other less conventional liver allografts.

Data supporting inferior outcomes with EAD have largely been based on outcomes pertaining to conventional DBD grafts [11,12,13]. These studies have shown inferior patient and graft survival as well as increased resource utilization such as ventilator days, occurrence of renal failure, and longer hospital length of stay [11,12,13,14,15]. The rising prevalence of EAD with the use of nonconventional grafts, such as those from DCD donors, is arguably expected; however, increasingly favorable outcomes have also been demonstrated with such donors despite the presence of at-risk variables [4,5,18,24,25,26]. In this series, 31.8% of grafts were from DCD donors and post-transplant outcomes were similar in regard to resource utilization (ICU and hospital length of stay), prevalence of AKI, and patient and graft survival. Mazilescu et al. reported similar findings where EAD was observed in 71% of DCD grafts and had satisfactory patient and graft survival [23]. Similarly, Croome et al. observed a 68.4% prevalence of EAD in DCD grafts and lack of impact on outcomes [17]. For DCD grafts, the mechanism of donation requiring cessation of donor circulation, presence of warm ischemia, and associated hepatocyte injury are likely important variables responsible for EAD.

Recipients in Group 2 had a lower MELD score, were more likely to have malignancy as an indication for transplant, and were less likely to be undergoing re-transplantation. Despite seemingly favorable characteristics, Group 2 recipients were more likely to experience EAD at the time of transplant as a result of donor-related variables. Group 2 grafts had a higher DRI score and were more likely to come from DCD donors. They were also more likely to be declined by local centers, thereby resulting in regional sharing and higher CITs. In contrast, Group 1 recipients had a higher MELD score, were more likely to have indications for transplant other than malignancy, and were more likely to be undergoing re-transplantation. Although these characteristics can contribute to increased surgical complexity, Group 1 recipients were less likely to experience EAD following transplant as a result of more favorable donor and graft characteristics. This juxtaposition highlights the importance of optimal donor—recipient pairing, particularly when utilizing less ideal grafts. Within the United Sates, there is considerable variability in liver transplantation rates and the median MELD at the time of transplant. Some attribute this finding to deceased donor availability; however, much of this variability is explained by transplant center practices and willingness to use ideal, standard, and non-ideal grafts, such as those coming from DCD donors [25]. Although our center performs a high volume of liver transplants at a low median MELD, these opportunities arise as a result of organ acceptance practices. It is frequently assumed that LT in lower MELD patients is technically easier; however, these patients often have significant liver-related complications that are underrepresented by their MELD score. Moreover, because these patients are disadvantaged on the waitlist as a result of their MELD, they are more likely to receive less ideal grafts that are predisposed to EAD. This recipient—donor combination can be challenging due to significant post-reperfusion coagulopathy. Proactive management of early coagulopathy and expectant management of lab abnormalities are important aspects that allow for the successful utilization of these grafts.

Although a lack of impact of EAD in DCD LT outcomes has been inferred, inferior outcomes with EAD in DBD grafts continue to be commonly reported [17,23]. Given the broader graft diversity in this current study cohort, which was inclusive of standard DBD grafts as well as those with moderate steatosis and increased CIT (regional/national share and post-cross clamp allocation), we sought to reassess this viewpoint. Donor—recipient pairing, along with selective utilization of nonconventional grafts, such as those with moderate steatosis, is a critical component guiding outcomes. EAD was only observed in 40.1% of DBD grafts compared to 79.4% of DCD grafts. Higher donor BMI, moderate steatosis on biopsy, and higher CITs were more common in DBD grafts with EAD, suggesting differing risk variables as contributors to EAD compared to DCD grafts. Unlike prior reports, we did not observe increased lengths of stay or differences in patient or graft survival for DBD grafts with EAD. PNF events were overall low; however, these all occurred in DBD grafts. Higher donor BMI and CIT, as well as abnormal biopsy findings were observed in these cases, suggesting that additional caution should perhaps be taken in the context of such variables. Growing awareness, individualized cause recognition, and adjustments in the management of EAD are likely significant contributors to the increasingly successful use of nonconventional donors.

It is noteworthy that the majority of EAD grafts in this study met only one EAD criterion, AST > 2000 U/L during the first week post-LT. Fodor et al. previously noted that bilirubin and INR have a stronger predictive capacity for patient and graft survival compared to AST [26]. Similarly, Wang et al. also noted that EAD outcomes varied by the number of observed criteria, with outcomes of recipients having one criterion being superior to those with two or three criteria [22]. These observations likely remain valid for standard conventional grafts where laboratory abnormalities post-LT remain uncommon. For nonconventional donors, such as grafts from DCD donors, abnormal laboratory values are exceedingly common in the early period following LT and can be managed with supportive treatment [4].

To our knowledge, this is the first study investigating EAD that has specifically focused on nonconventional donor types. It is also one of only a few studies to show a lack of impact of EAD on DCD and DBD post-transplant outcomes, including length of stay, AKI, and patient and graft survival. We do, however, acknowledge that, as a single center study, there are inherent limitations in generalizability due to practice-specific nuances. As a center with experience utilizing nonconventional grafts, the outcomes described here are reflective of carefully selected grafts. Donor—recipient pairing is a crucial component influencing outcomes, and similar outcomes would likely not be observed with the use of the same grafts in higher MELD patients. In addition, with the increased use of normothermic machine perfusion in liver transplantation, the clinical phenomenon of EAD will likely become increasingly uncommon, even with increased risk grafts. We hope that by sharing our experience, we can help improve the utilization of nonconventional donors where EAD is a common and non-detrimental occurrence.

## 5. Conclusions

In summary, the ongoing reassessment of EAD, risk factors for EAD, as well as graft and patient outcomes is important to allow for further expansion of the donor pool and the successful use of non-ideal grafts. With efforts to expand the donor pool, EAD has become commonplace and no longer portends inferior outcomes, particularly for DCD grafts. In the context of non-ideal grafts, EAD should be managed supportively and should not be viewed as a deterrent to utilization.

## Figures and Tables

**Figure 1 medicina-58-00821-f001:**
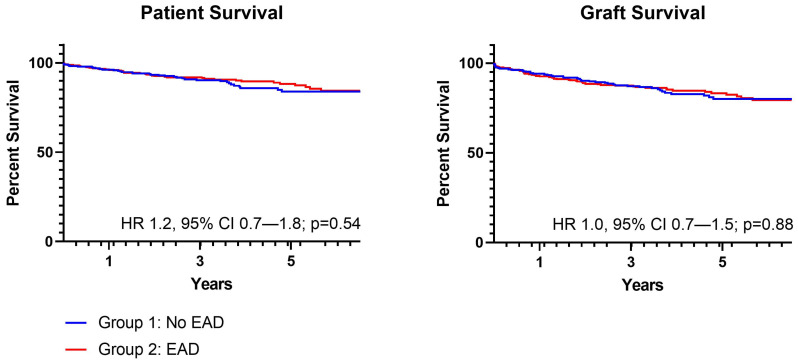
Patient and graft survival.

**Table 1 medicina-58-00821-t001:** Recipient and donor demographics.

	Group 1: No EAD (*n* = 290)	Group 2: EAD (*n* = 321)	*p* Value
**Recipient**			
Age (years)	56.4 ± 10.2 (58.0)	57.2 ± 10.5 (59.0)	0.32
Female	108 (37.2%)	102 (31.8%)	0.16
Hispanic	54 (18.6%)	43 (13.4%)	0.08
Race - White - Black - Other	222 (76.6%) 0 (0.0%) 14 (4.8%)	258 (80.4%) 7 (2.2%) 13 (4.0%)	0.04
MELD	23 (17, 30)	21 (15, 26)	0.0002
Diagnosis - Non-cholestatic - Metabolic - Malignant - Cholestatic - Other	126 (43.4%) 67 (23.1%) 40 (13.8%) 22 (7.6%) 35 (12.1%)	148 (46.1%) 59 (18.4%) 68 (21.2%) 29 (9.0%) 17 (5.3%)	0.004
PVT	43 (14.8%)	61 (19.0%)	0.17
TIPS	27 (9.3%)	41 (12.8%)	0.17
Re-Transplant	24 (8.3%)	9 (2.8%)	0.003
**Donor**			
Age (years)	48.0 ± 18.1 (50.0)	48.5 ± 13.0 (52.0)	0.65
DRI	1.6 ± 0.5 (1.6)	1.9 ± 0.5 (1.9)	<0.0001
BMI (kg/m^2^)	28.4 ± 7.4 (26.8)	31.0 ± 8.3 (30.0)	<0.0001
Sharing - Local - Regional - National	103 (35.5%) 137 (47.2%) 50 (17.2%)	65 (20.2%) 198 (61.7%) 58 (18.1%)	<0.0001
DCD donor	40 (13.8%)	154 (48.0%)	<0.0001
DCD dWIT (minutes)	23.0 (16.0, 26.0)	21.0 (18.0, 25.0)	0.53
CIT (hours)	5.9 ± 1.9 (5.5)	6.4 ± 1.9 (6.0)	0.001

**Table 2 medicina-58-00821-t002:** Post-transplant outcomes.

	Group 1: No EAD (*n* = 290)	Group 2: EAD (*n* = 321)	*p* Value
ICU LOS (days)	1.0	2.0	0.60
Hospital LOS (days)	5.0	6.0	0.24
EAD - 1 criterion - 2 criteria - 3 criteria		286 (89.1%) 28 (8.7%) 7 (2.2%)	---
	---		
Peak AST (U/L)	913 (554, 7700)	4010 (2853, 7700)	<0.0001
AKI	117 (40.3)	140 (43.6%)	0.41
PNF	1 (0.3%)	4 (1.3%)	0.22
Early Allograft Dysfunction - T Bili > 10 on day 7 - INR > 1.6 day 7 - AST > 2000 in first week		15 (4.7%) 21 (6.5%) 321 (100.0%)	---
	---		

**Table 3 medicina-58-00821-t003:** Post-transplant laboratory values.

	Group 1: No EAD	Group 2: EAD	*p* Value
1 Week Post-LT - T Bili - ALT - AST - Alk Phos	1.5 (1.0, 2.5) 115.0 (77.8, 190.3) 41.0 (29.0, 62.3) 140.5 (94.0, 211.3)	1.9 (1.1, 4.5) 229.0 (155.0, 338.0) 55.0 (38.0, 81.0) 167.0 (117.0, 257.0)	<0.0001 <0.0001 0.03 0.01
6 Weeks Post-LT - T Bili - ALT - AST - Alk Phos	0.6 (0.4, 0.8) 19.0 (13.0, 31.5) 21.0 (16.0, 27.0) 100.0 (79.0, 143.0)	0.6 (0.4, 1.1) 26.0 (16.0, 45.0) 22.0 (17.0, 32.0) 125.0 (87.0, 198.0)	0.004 0.05 0.001 <0.0001
1 Year Post-LT - T Bili - ALT - AST - Alk Phos	0.6 (0.3, 0.7) 23.0 (17.0, 38.0) 27.0 (20.0, 34.0) 114.0 (85.0, 164.8)	0.5 (0.3, 0.7) 23.0 (16.0, 38.0) 25.0 (21.0, 33.0) 113.0 (88.0, 156.0)	0.29 0.42 0.18 0.08

**Table 4 medicina-58-00821-t004:** Subgroup analysis comparing EAD in DCD and DBD grafts.

	Group A: EAD DCD (*n* = 154)	Group B: EAD DBD (*n* = 167)	Group C: No EAD DCD (*n* = 40)	Group D: No EAD DBD (*n* = 250)	*p* Value
**Recipient**					
Age (years)	58.5 ± 9.1	56.0 ± 11.5	58.1 ± 9.4	56.1 ± 10.4	0.08
MELD (median)	20.5 ± 6.0 (22.0)	21.0 ± 9.0 (21.0)	21.5 ± 6.1 (23.0)	23.6 ± 9.6 (23.0)	0.001
Diagnosis - Non-cholestatic - Metabolic - Malignant - Cholestatic - Other	73 (47.4%) 41 (26.6%) 29 (18.8%) 10 (6.5%) 1 (0.6%)	75 (44.9%) 18 (10.8%) 39 (23.4%) 19 (11.4%) 16 (9.6%)	17 (42.5%) 12 (30.0%) 7 (17.5%) 1 (2.5%) 3 (7.5%)	110 (44.0%) 55 (22.0%) 33 (13.2%) 21 (8.4%) 31 (12.4%)	0.001
Re-Transplant	---	9 (5.4%)	---	24 (9.6%)	0.12
**Donor**					
Age (years)	49.2 ± 11.4 (51.5)	47.8 ± 18.9	49.6 ± 15.2 (53.0)	47.6 ± 15.2 (50.0)	0.62
DRI	2.3 ± 0.4 (2.3)	1.6 ± 0.4 (1.5)	2.3 ± 0.3 (2.3)	1.5 ± 0.4 (1.5)	<0.0001
BMI (kg/m^2^)	20.5 ± 6.0 (21.5)	32.2 ± 8.5 (31.5)	26.3 ± 5.2 (25.9)	28.7 ± 7.6 (27.2)	<0.0001
Sharing - Local - Regional - National	27 (17.5%) 106 (68.8%) 21 (13.6%)	38 (22.7%) 92 (55.1%) 37 (22.2%)	10 (25.0%) 23 (57.5%) 7 (17.5%)	93 (37.2%) 114 (45.6%) 43 (17.2%)	<0.0001
CIT (hours)	5.6 ± 1.2 (5.4)	7.2 ± 2.1 (6.9)	5.5 ± 1.0 (5.4)	6.0 ± 2.0 (5.5)	<0.0001
**Post-LT Outcomes**					
ICU LOS (days)	2.0 (1.0, 2.0)	1.0 (1.0, 2.0)	2.0 (1.0, 3.0)	1.0 (1.0, 2.0)	0.37
Hospital LOS (days)	6.0 (5.0, 8.0)	6.0 (5.0, 9.0)	5.0 (4.0, 8.0)	5.0 (4.0, 8.0)	0.12
PNF	0 (0.0%)	4 (2.4%)	1 (2.5%)	0 (0.0%)	0.02

**Table 5 medicina-58-00821-t005:** Characteristics of grafts with EAD and PNF.

PNF Case	MELD	Donor Age (Years)	BMI (kg/m^2^)	Allocation	DRI	Post-Cross Clamp Offer	CIT (Hours)	Liver Allograft Biopsy
**#1**	14	49	42.1	National	1.4	Yes	8.1	Steatosis, 20–30%
**#2**	32	57	26.2	Local	1.4	No	4.1	Severe arteriolar hyaline
**#3**	8	63	24.2	National	1.9	Yes	7.7	>70% hepatocellular necrosis
**#4**	39	36	36.1	Regional	1.2	No	6.6	Normal biopsy

## Data Availability

The data that support the findings of this study are available from the corresponding author upon reasonable request.

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
