# Peer review of "Decreasing Significance of Early Allograft Dysfunction with Rising Use of Nonconventional Donors"

_medicina, 2022, doi:10.3390/medicina58060821_

Round 1

Reviewer 1 Report

This research targets an important topic in the field of liver transplantation; which is early allograft dysfunction and its impact on early(1-6weeks) and late (1 year) post transplant outcome.

The authors did a great work but, few points should be revised:

The research is a retrospective study of the "prevalence" (not incidence) of EAD in a cohort of liver transplant recipients. So, this should be corrected in the abstract, aim of work and also in the methodology sections.

What is meant by "updated" cohort? mentioned in the abstract and in line 60. 

The research is mainly comparing patients with and without EAD. So, why not state that there were 2 groups of recipients; G1 and G2. It is easier to follow while reading the manuscript.

What is the situation with re-transplantation. Were there any re-transplanted patients included in the analysis?. Was retransplantation one of the exclusion criteria?. Are the 33patients included in Table 1 were originally re-transplanted patients or recipients needing re-transplantation.? If so; why not including retransplantation as one of the outcomes.?

Line 101: "Recipients of grafts with EAD had lower MELD scores and were less likely to be re-transplants" The sentence is confusing. Does it mean that recipients in the EAD group (say G1) had lower MELD scores than recipients in the No EAD group (say G2)?. Is there an explanation for this?.

There should be a mention of the definition of primary graft  nonfunction (PNF). 

No mention of mortality in the outcome!. Even if there is no mortality, it should be mentioned.

Author Response

June 6, 2022

Dear Reviewer:

Thank you for giving us the opportunity to revise and resubmit our manuscript. We have tried to answer all of your questions and have updated our paper to reflect your recommendations. With your guidance, we think these updates have improved the quality and impact of the paper and we hope that our changes are fitting with your expectations. All of the authors have reviewed and approved of this revision.

Caroline C. Jadlowiec, M.D.

Reviewer 1

This research targets an important topic in the field of liver transplantation; which is early allograft dysfunction and its impact on early(1-6weeks) and late (1 year) post-transplant outcome.

The authors did a great work but, few points should be revised:

  1. The research is a retrospective study of the "prevalence" (not incidence) of EAD in a cohort of liver transplant recipients. So, this should be corrected in the abstract, aim of work and also in the methodology sections.

We have made this correction throughout the manuscript.

  1. What is meant by "updated" cohort? mentioned in the abstract and in line 60.

The phrase “updated cohort” was meant to highlight the use of both conventional DBD grafts as well as less conventional grafts, such as those coming from DCD donors. Within our study, 31.8% of grafts came from DCD donors, 17.7% of grafts were nationally allocated indicating local-regional decline, 8.7% of grafts had moderate steatosis and 16.4% were accepted post-cross clamp of which 57 were DCD post-cross clamp grafts. To date, much of the published literature on EAD and outcomes has been based on conventional DBD grafts, aside from two studies that included DCD grafts. For improved clarity, we have updated this sentence to read, “an updated cohort inclusive of DBD, DCD, and other less conventional grafts.”

  1. The research is mainly comparing patients with and without EAD. So, why not state that there were 2 groups of recipients; G1 and G2. It is easier to follow while reading the manuscript.

Thank you for the suggestion. We have made this change throughout.

  1. What is the situation with re-transplantation. Were there any re-transplanted patients included in the analysis? Was retransplantation one of the exclusion criteria? Are the 33patients included in Table 1 were originally re-transplanted patients or recipients needing re-transplantation.? If so; why not including retransplantation as one of the outcomes.?

We included recipients undergoing re-transplantation to better illustrate a sense of the case mix to the readers. Re-transplantation, portal vein thrombosis and TIPS are considered, by some, to be markers of increased case complexity. Many transplant centers avoid using less ideal grafts in the context of these variables. The re-transplant recipients included in this study received their original transplants at an earlier time.

  1. Line 101: "Recipients of grafts with EAD had lower MELD scores and were less likely to be re-transplants" The sentence is confusing. Does it mean that recipients in the EAD group (say G1) had lower MELD scores than recipients in the No EAD group (say G2)? Is there an explanation for this?

This is correct. Recipients in Group 2 (grafts with EAD) had a lower MELD score and were less likely to have been previously transplanted. Within the United Sates, there is considerable variability in liver transplantation rates and the median MELD at the time of transplant. Some attribute this to deceased donor availability; however, much of this variability is explained by transplant center practices and willingness to use ideal, standard and non-ideal grafts, such as those coming from DCD donors. Although our center performs a high volume of transplants at a low median MELD, these opportunities arise not because of a surplus of donors, but rather as a result of organ acceptance practices. It is frequently assumed that liver transplantation in lower MELD patients is easy, however this is not necessarily the case. Recipients listed with their biologic MELD often have significant complications related to their liver disease that are underrepresented by their MELD score. Moreover, because these patients are disadvantaged on the waitlist as a result of their MELD, they are more likely to receive non-deal grafts, e.g. post-allocation (cross-clamp) grafts, DCD grafts, regionally allocated grafts (due to local decline) and grafts with moderate steatosis. This recipient-donor combination can be challenging particularly in the setting of significant post-reperfusion coagulopathy and EAD. As a result, Group 2 recipients had a lower MELD score, were less likely to be undergoing re-transplantation and were more likely to receive regionally allocated DCD grafts with a higher DRI score. We have modified the language in the original sentence to say “undergoing re-transplantation” rather than “be re-transplants” to add clarity.

  1. There should be a mention of the definition of primary graft nonfunction (PNF).

We originally included a brief definition of PNF in the methods section (lines 81-82). PNF was defined as early graft failure, occurring within 7 days of liver transplantation, in the absence of technical and immunologic issues. Our definition also aligns with the Organ Procurement and Transplantation Network (OPTN) listing criteria [AST greater than or equal to 3,000 U/L and at least one of the following: 1) INR >2.5, 2) arterial pH less <7.30, 3), venous pH <7.25, 4) lactate >4 mmol/L] apart from the single PNF event that occurred in the no EAD cohort. That graft came from a small donor and did not have AST >3000, however this was likely due to graft size and overall lower hepatocyte mass. The recipient otherwise met criteria for PNF based on lactate, pH and absence of technical and immunologic factors.

Reference

  1. OPTN Policies. https://optn.transplant.hrsa.gov/media/eavh5bf3/optn_policies.pdf; Published April 28, 2022. Accessed June 3, 2022.

  1. No mention of mortality in the outcome! Even if there is no mortality, it should be mentioned.

Our apologies if this wasn’t clear however, we included data on patient and graft survival in the outcomes (lines 146-150) and Figure 1.

There were no differences in patient (HR 1.2, 95% CI 0.7-1.8; p=0.54) or graft (HR 1.0, 95% CI 0.7-1.5; p=0.88) survival (Figure 1).  Median follow-up was 3.6 years (IQR 2.4-5.1) in Group 1 and 4.1 years (IQR 2.7-5.6) for Group 2. One-year patient survival for grafts with and without EAD was 96.0% and 94.1%; 1-year graft survival was 92.5% and 92.1%.

               We also included this data in the abstract and the conclusion.

Reviewer 2 Report

The authors have written an original paper studying the role of using Early Allograft Dysfunction criteria for nonconventional lover donors. The paper is original, valuable in concept, and shows great promise. There are, however, several issues to be addressed:

- EAD criteria are defined twice, once in introduction and again in the materials & methods. Please correct this.

- the paper is rich in acronyms and some conventions should be adhered to when using them: definition on the first use (some are not defined, others are defined more than once), usage of plurals or integrating grammar (i.e. "We analyzed 611 LT at our center" should read "611 LTs" or "611 LT patients"), avoiding usage at the beginning of the sentence where possible (CIT), and using it consistently throughout (LT is, at times, written in the long form). Also, some acronyms are defined in the abstract but never used again, which makes no sense (AKI, IC...).

-  multiple grammar mistakes - please check all text carefully (e.g. "6 weeks of one-year", "defined by elevation bilirubin", "Despite \ data ", and so on).

- Figure 1 needs a caption.

- lines 168-173 are just repeating results and have no place in the Discussions chapter.

- all the classes of patient demographics should be shortly addressed in the discussions chapter, otherwise the presence of this data in the table is futile.

- further limitations of this study should be sought and described in the last paragraph of the discussions, alongside the mentioned "single center study".

- the last sentence of the conclusions should be rewritten to increase clarity.

- were other predictors of graft failure and patient survival used in the management of these patients? how did EAD scores correlate with those? can we draw up a ROC curve in respect to a gold standard?

- a short paragraph describing the software and statistics employed in this study should be placed in the last part of Materials and Methods, including the tests used and testing the data distribution/normality.

- further discussions should be made regarding the lack of statistical difference in survival of the patients and grafts in the two sublots, as there were many confounding variables including liver failure cause, race, donor risk, BMI and many others.

Respectfully submitted

Author Response

June 7, 2022

Dear Reviewer:

Thank you for giving us the opportunity to revise and resubmit our manuscript. We have tried to answer all of your questions and have updated our paper to reflect your recommendations. With your guidance, we think these updates have improved the quality and impact of the paper and we hope that our changes are fitting with your expectations. All of the authors have reviewed and approved of this revision.

Caroline C. Jadlowiec, M.D.

Reviewer 2

The authors have written an original paper studying the role of using Early Allograft Dysfunction criteria for nonconventional liver donors. The paper is original, valuable in concept, and shows great promise. There are, however, several issues to be addressed:

  1. EAD criteria are defined twice, once in introduction and again in the materials & methods. Please correct this.

We have removed the definition from the introduction, and this remains available in the methods section. 

  1. The paper is rich in acronyms and some conventions should be adhered to when using them: definition on the first use (some are not defined, others are defined more than once), usage of plurals or integrating grammar (i.e. "We analyzed 611 LT at our center" should read "611 LTs" or "611 LT patients"), avoiding usage at the beginning of the sentence where possible (CIT), and using it consistently throughout (LT is, at times, written in the long form). Also, some acronyms are defined in the abstract but never used again, which makes no sense (AKI, IC...).

That is a fair observation. We have reviewed the manuscript closely and have made changes consistent with the above feedback.

  1. Multiple grammar mistakes - please check all text carefully (e.g. "6 weeks of one-year", "defined by elevation bilirubin", "Despite \ data ", and so on).

Our apologies. We have taken the opportunity to review the manuscript and make corrections.

  1. Figure 1 needs a caption.

This has been added.

  1. Lines 168-173 are just repeating results and have no place in the Discussions chapter.

We have removed this paragraph.

  1. All the classes of patient demographics should be shortly addressed in the discussions chapter, otherwise the presence of this data in the table is futile.

Thank you for this suggestion. This question also aligns with question 5 from Reviewer 1. Recipients in Group 2 (grafts with EAD) had a lower MELD score, were more likely to have malignancy as an indication for transplant and were less likely to be undergoing re-transplantation. Despite seemingly favorable characteristics, Group 2 recipients were more likely to experience EAD at the time transplant as a result of donor-related variables. Group 2 grafts had a higher DRI score and were more likely to come from DCD donors. They were also more likely to be declined by local centers, thereby resulting in regional sharing, and higher CITs. By contrast, Group 1 recipients had a higher MELD score, were more likely to have indications for transplant other than malignancy and were more likely to be undergoing re-transplantation. Although these characteristics can contribute to increased surgical complexity, Group 1 recipients were less likely to experience EAD following transplant, likely as a result of more favorable donor and graft characteristics such as lower DRI score, DBD status and shorter CIT.  This juxtaposition highlights the importance of optimal donor-recipient pairing, particularly when utilizing less ideal grafts. Within the United Sates, there is considerable variability in liver transplantation rates and the median MELD at the time of transplant. Some attribute this finding to deceased donor availability, however, much of this variability is explained by transplant center practices and willingness to use ideal, standard and non-ideal grafts, such as those from DCD donors. Although our center performs a high volume of transplants at a low median MELD, these opportunities arise not because of a surplus of donors, but rather as a result of organ acceptance practices. It is frequently assumed that liver transplantation in lower MELD patients is technically easier; however, low MELD recipients listed for transplant often have significant complications related to their liver disease that are underrepresented by their MELD score. Moreover, because these patients are disadvantaged on the waitlist as a result of their MELD, they are more likely to receive less ideal grafts that are predisposed to EAD. This recipient-donor combination can be challenging due to significant post-reperfusion coagulopathy. Proactive management of early coagulopathy and expectant management of lab abnormalities are important aspects for success.  We have added this into our discussion.

  1. Further limitations of this study should be sought and described in the last paragraph of the discussions, alongside the mentioned "single center study".

We agree and have added to the discussion. As a center with experience in utilization of nonconventional grafts, the outcomes described here are reflective of carefully selected grafts. Donor-recipient pairing is a crucial component influencing outcomes, and similar outcomes would likely not be observed with use of the same grafts in higher MELD patients. 

  1. The last sentence of the conclusions should be rewritten to increase clarity.

This has been re-written.

  1. Were other predictors of graft failure and patient survival used in the management of these patients? how did EAD scores correlate with those? can we draw up a ROC curve in respect to a gold standard?

Thank you for this suggestion. We felt that this question was better answered through multivariate analysis which we have added. Multivariable regression analysis was completed looking at the risk of graft loss at 1-year post-LT. In a multivariate model, after accounting for differences in graft type and CIT, EAD was not associated with graft loss at 1-year post-LT (OR 1.1, 95% CI 0.6-2.1).

  1. A short paragraph describing the software and statistics employed in this study should be placed in the last part of Materials and Methods, including the tests used and testing the data distribution/normality.

We apologize for this omission. We have added this to our methods section.

  1. Further discussions should be made regarding the lack of statistical difference in survival of the patients and grafts in the two sublots, as there were many confounding variables including liver failure cause, race, donor risk, BMI and many others.

We agree. Thank you for this suggestion. We have organized the subgroup analysis data into table form (Table 4) and have expanded its contents. Donor-recipient pairing, along with selective utilization of nonconventional grafts, such as those with moderate steatosis, is a critical component guiding outcomes.  EAD was observed in only 40.1% of DBD grafts compared to 79.4% of DCD grafts. Higher donor BMI, moderate steatosis on biopsy and higher CITs were more common in DBD grafts with EAD suggesting differing risk variables as contributors to EAD compared to DCD grafts. Unlike prior reports, we did not observe increased lengths of stay or differences in patient or graft survival for DBD grafts with EAD. PNF events were overall low, however it should be noted that these all occurred with DBD grafts. Higher donor BMI and CIT as well as abnormal biopsy findings were observed in these cases suggesting that additional caution perhaps be taken in the context of such variables (Table 5). We have further updated this in our discussion.

Table 4. Subgroup Analysis comparing EAD in DCD and DBD Grafts

Group A:

EAD DCD

(n=154)

Group B:

EAD DBD

(n=167)

Group C:

No EAD DCD

(n=40)

Group D:

No EAD DBD

(n=250)

P Value

Recipient

Age (years)

58.5±9.1

56.0±11.5

58.1±9.4

56.1±10.4

0.08

MELD (median)

20.5±6.0 (22.0)

21.0±9.0 (21.0)

21.5±6.1 (23.0)

23.6±9.6 (23.0)

0.001

Diagnosis

-                   Non-cholestatic

-                   Metabolic

-                   Malignant

-                   Cholestatic

-                   Other

73 (47.4%)

41 (26.6%)

29 (18.8%)

10 (6.5%)

1 (0.6%)

75 (44.9%)

18 (10.8%)

39 (23.4%)

19 (11.4%)
16 (9.6%)

17 (42.5%)

12 (30.0%)

7 (17.5%)

1 (2.5%)

3 (7.5%)

110 (44.0%)

55 (22.0%)

33 (13.2%)

21 (8.4%)

31 (12.4%)

0.001

Re-Transplant

---

9 (5.4%)

---

24 (9.6%)

0.12

Donor

Age (years)

49.2±11.4 (51.5)

47.8±18.9

49.6±15.2 (53.0)

47.6±15.2 (50.0)

0.62

DRI

2.3±0.4 (2.3)

1.6±0.4 (1.5)

2.3±0.3 (2.3)

1.5±0.4 (1.5)

<0.0001

BMI (kg/m2)

20.5±6.0 (21.5)

32.2±8.5 (31.5)

26.3±5.2 (25.9)

28.7±7.6 (27.2)

<0.0001

Sharing

-                   Local

-                   Regional

-                   National

27 (17.5%)

106 (68.8%)

21 (13.6%)

38 (22.7%)

92 (55.1%)

37 (22.2%)

10 (25.0%)

23 (57.5%)

7 (17.5%)

93 (37.2%)

114 (45.6%)

43 (17.2%)

<0.0001

CIT (hours)

5.5±1.0 (5.4)

7.2±2.1 (6.9)

5.5±1.0 (5.4)

6.0±2.0 (5.5)

<0.0001

Post-LT Outcomes

ICU LOS (days)

2.0 (1.0, 2.0)

1.0 (1.0, 2.0)

2.0 (1.0, 3.0)

1.0 (1.0, 2.0)

0.37

Hospital LOS (days)

6.0 (5.0, 8.0)

6.0 (5.0, 9.0)

5.0 (4.0, 8.0)

5.0 (4.0, 8.0)

0.12

PNF

0 (0.0%)

4 (2.4%)

1 (2.5%)

0 (0.0%)

0.02

Table 5. Characteristics of Grafts with EAD and PNF

PNF

case

MELD

Donor age

(years)

BMI

(kg/m2)

Allocation

DRI

Post-cross clamp offer

CIT

(hours)

Liver allograft biopsy

#1

14

49

42.1

National

1.4

Yes

8.1

Steatosis, 20-30%

#2

32

57

26.2

Local

1.4

No

4.1

Severe arteriolar hyaline

#3

8

63

24.2

National

1.9

Yes

7.7

>70% hepatocellular necrosis

#4

39

36

36.1

Regional

1.2

No

6.6

Normal biopsy

Round 2

Reviewer 2 Report

The authors have performed all suggested and required corrections.